# Gender differences in unpaid care work and psychological distress in the UK Covid-19 lockdown

**Baowen Xue**⊙*[☯], **Anne McMunn**[☯]

Department of Epidemiology & Public Health, UCL, London, United Kingdom

[☯] These authors contributed equally to this work.

* baowen.xue.10@ucl.ac.uk

## Abstract

### Objective

To describe how men and women divided childcare and housework demands during the height of the first Covid-19 lockdown in the UK, and whether these divisions were associated with worsening mental health during the pandemic.

### Background

School closures and homeworking during the Covid-19 crisis have resulted in an immediate increase in unpaid care work, which draws new attention to gender inequality in divisions of unpaid care work.

### Methods

Data come from the wave 9 (2017–19) of Understanding Society and the following April (n = 15,426) and May (n = 14,150) waves of Understanding Society Covid-19 study. Psychological distress was measured using the General Health Questionnaire (GHQ) at both before and during the lockdown, and unpaid care work was measured during the lockdown. Linear regression models were used.

### Results

Women spent much more time on unpaid care work than men during lockdown, and it was more likely to be the mother than the father who reduced working hours or changed employment schedules due to increased time on childcare. Women who spent long hours on housework and childcare were more likely to report increased levels of psychological distress. Working parents who adapted their work patterns increased more psychological distress than those who did not. This association was much stronger if he or she was the only member in the household who adapted their work patterns, or if she was a lone mother. Fathers increased more psychological distress if they reduced work hours but she did not, compared to neither reducing work hours.

**Data Availability Statement:** We use the Understanding Society data in this study. Understanding Society data are publicly available to researchers through the UK Data Service. Researchers who would like to use Understanding

Society can find more information here: https://www.understandingsociety.ac.uk/documentation/access-data.

**Funding:** AM has been supported by the Economic and Social Research Council International Centre for Lifecourse Studies in Society and Health (ICLS) [grant number ES/J019119/1]. https://esrc.ukri.org/ The funders had no role in study design, data collection and analysis, decision to publish, or preparation of the manuscript.

**Competing interests:** The authors have declared that no competing interests exist.

## Conclusion

There are continued gender inequalities in divisions of unpaid care work. Juggling home working with homeschooling and childcare as well as extra housework is likely to lead to poor mental health for people with families, particularly for lone mothers.

## Introduction

Amongst contemporary couples in the UK, women continue to spend more time than men doing unpaid care work. There has been concern that the shutdowns and school closures during lockdown may exacerbate existing inequalities between men and women and between couple parents and lone parents in terms of mental health; yet, there have been no empirical analyses on this. The aim of this work is to answer how men and women in the UK divided childcare and housework during lockdown, and whether this is associated with changes in levels of psychological distress.

In March 2020, childcare facilities and schools in the UK were shut down in response to the ongoing Covid-19 pandemic, and all but keyworkers were also told not to go into work and to work from home if they could (also known as the first lockdown in the UK). Many schools carried on remote homeschooling in April and May and schools and childcare facilities did not begin to re-open until June. This nationwide lockdown signified severe restrictions on social contact and a fairly immediate increase in unpaid care work, particularly for families with young children [1–4]. Unpaid care work is defined as all unpaid services provided within a household for its members, including care of persons and housework [5].

Combining increased levels of unpaid care work with remote working from home during the lockdown may have increased feelings of psychological distress through reduced time for sleep and leisure, and the stress of trying to meet competing demands [6]. Previous research has documented the negative mental health effects of long working hours [7, 8] and working non-standard hours [9, 10], as well as combining care provision with full-time employment [11]. Role strain theory has been widely used to explain how the potentially competing demands of unpaid care work and employment responsibilities may be related to mental health [12]. According to this theory, human energy is limited, and *role overload* occurs when demands on energy and stamina exceed the individual's capacity. The role strain theory suggests that the more demands within a role, or the more roles a person occupies, the more role strain experienced and the greater the likelihood of negative effects on mental health [13]. Thus, long hours spent in housework and childcare may lead to role overload, particularly if combined with paid work, resulting in role conflict and subsequent psychological distress. In addition, work and family may intrude on one another with one domain spilling over onto the other. The *spillover* process may involve stressors and the transmission of attitudes, emotions, beliefs, and behaviours from one domain to another, as well as in multiple domains on the same day [14]. Negative spillover between work and family, which is most frequently characterized by various types of work-family conflict or interference, has been linked with psychological distress and marital dissatisfaction [15]. However, positive spillover between work and family, such as having a supportive partner, is positively related both to well-being at work and general well-being [16]. The lockdown brings the work and home activities together as never before; thus work-family spillover may be more obvious and immediate than prior to the pandemic, which may directly influence on health and well-being.

Moreover, previous research on family and well-being has suggested the crossover effects of the partner, that is one's well-being may not only depend on one's own but also on their

partners' involvement in work and family [17, 18]. Perceptions of equality and reciprocity within couples are likely to be important [13, 19], so how couples divide the unpaid care work and how they make an adjustment on paid work due to unpaid care may contribute to couples' perceptions of equity (or inequity) in a relationship, and thus potentially influence health and well-being.

Prior to the pandemic, recent evidence suggested that, amongst contemporary couples in the UK, women continue to spend more time than men doing housework, childcare and caring for adults [20] and initial survey data in the UK suggested that the increased domestic workload during lockdown has fallen more to women than to men [21]. In the short-term, juggling home working with homeschooling and childcare is likely to lead to long-hour days and working non-standard schedules for many parents. In the longer-term, increased unpaid care work may have implications for employment participation, pay and progression, exacerbating the persistent gender pay gap [22].

A national survey of the UK during lockdown also showed that women's psychological distress rose more than men's during lockdown [23]. However, whether and how women and men's increased psychological distress during lockdown can be explained by unpaid care work has not yet been analysed, and a couple perspective in the division of unpaid care work is needed when analysing the potential association between unpaid care and mental health during lockdown.

In addition, lone parents, the vast majority of whom are women, are at a greater risk of poverty and poor health [24]. Without the support of another adult in the household to balance childcare arrangements, many lone parents struggled to combine work and family responsibilities even before the pandemic [25]. The practicalities of self-isolation and school and nursery closures during lockdown may exacerbate existing inequalities between couple parents and lone parents, which need to be explored when assessing the impact of Covid-19 lockdown on mental health.

Using the unique household design of the UK Household Longitudinal Study data collections during April and May 2020, we aim to describe gender divisions of unpaid care work during the height of the Covid-19 lockdown in the UK and its associations with psychological distress. Our research hypotheses are:

1. Women spent more time on unpaid care work than men during the first lockdown in the UK.

2. Within couples, women did a greater share of unpaid care work than men during lockdown.

3. Both high demands of unpaid care work per se and a high share of the division of unpaid care work within couples are associated with increased levels of psychological distress during lockdown.

4. These associations are stronger for women than for men.

5. For working parents, these associations are stronger for those who are not living with a partner than for those who are living with a partner.

## Materials and methods

### Data

Data for this study come from the Understanding Society Covid-19 study. *Understanding Society*, also known as UK Household Longitudinal Study, is a nationally representative

longitudinal study, which began in 2009 and recruited over 100,000 individuals in 40,000 households [26]. From April 2020, participants from the last two waves (wave 8 and wave 9) of Understanding Society were invited to complete a short web-survey to understand the experiences and reactions of the UK population to the Covid-19 pandemic. Participants complete one on-line questionnaire each month, which includes core content designed to track changes, alongside variable content adapted each month as the coronavirus situation develops. Our study uses the April (1st wave) and May (2nd wave) waves. A total of 17,452 respondents (6,166 are parents) answered the April wave and 14,811 respondents answered the May wave. The overall response rate of the April and May waves was 41.2% and 40.2%, respectively. Participants gave informed oral consent to take part in each wave of the study and were enrolled only after consent was provided. The survey procedures were approved by the Ethics Committee of University of Essex. Data is available to researchers via the UK Data Service. More details of the procedures can be found in the User Guide [27].

## Sample

This study involves several different unpaid care work exposures, and thus, we draw on six main sample types for our analysis: all participants, couples, parents, couple parents, working parents, and working couple parents. For investigating the number of hours individuals spend doing housework, we included all participants. To investigate couple divisions of housework, we restricted the sample to couples (wave 9 identifier was used to identify the members in a couple). To investigate the number of hours individuals spend providing childcare or homeschooling, we restricted the sample to parents, and we further limited our sample to parents who were living with a couple when analysing the division of childcare within couples. When analysing whether employment hours were reduced or adapted to accommodate childcare, our sample is limited to parents who were working before the pandemic, and we restrict the sample to working couple parents for couple level adjustment in employment. We conducted a 'complete case analysis', so participants with missing data on exposure, outcome or covariates were excluded. The process of sample selection was shown in S1 Table. For individual-level measures of unpaid care, the largest percentage of missing is from covariates which were measured in wave 9, as some participants did not participate in wave 9. For couple divisions of unpaid care, the largest percentage of missing is from the exposure variable (i.e., how couple divided the unpaid care), and this is because both members in a couple were excluded from the couple level analysis if one member was missing. The sample sizes for the six sample types were: all participants (April: 13218; May: 12472), couples (April: 7009; May: 5656), parents (April: 4174; May: 3179), couple parents (April: 1731; May: 1551), working parents (April: N/A; May: 2990), and working couple parents (April: N/A; May: 1572).

## Measures

**Unpaid care work.** Hours spent on doing housework in the last week and hours spent on childcare/homeschooling in the last week were measured in both April and May waves. Items asking whether employment hours were reduced or adapted because of the time spent doing childcare or homeschooling were added to the May wave on our request. In terms of couple-level exposures, women's share of involvement in housework or childcare was measured as the per cent of the total time that the couple spends on housework or childcare being done by women × 100. Whether employment hours were reduced or adapted due to childcare/homeschooling within couple parents were grouped into neither, both, mother only, or father only.

**Psychological distress.** Psychological distress was measured using the 12-item General Health Questionnaire (GHQ). GHQ is a validated scale of measuring non-psychotic

psychological distress and has been widely used in the community or non-clinical settings [28]. There are 12 items about respondents' depressive, anxiety symptoms, confidence and overall happiness, and each item has four response categories on a Likert scale (0 'less than usual', 1 'no more than usual', 2 'rather more than usual', and 3 'much more than usual'). Scores are summed and range from 0 (least distressed) to 36 (most distressed).

**Confounders.** In order to test the changes in levels of psychological distress, we adjusted for baseline GHQ scores, which was measured in wave 9 main survey. We have controlled for a number of socio-demographic characteristics. Participants' age was adjusted as a categorical variable (16/24, 25/34, 35/44, 45/54, 55/64, 65/74, 75+). We also controlled for ethnicity (White, Indian/ Pakistani/Bangladeshi/Chinese/Arab/any other Asian background, African/ Black/Caribbean, and other/mixed), whether living with a partner or not, and the number of children in the household (0, 1, 2+) by children's age group (aged 0–4, 5–15 and 16–18). Participants were asked to recall their working hours before the pandemic (in January or February 2020), so baseline working hour was adjusted as a categorical variable (not working, working part-time which is <30 h/w, working full-time which is 30 to 40h/w, and working full-time with long hours which is higher than 40h/w). Information on educational qualification and occupational class were not collected in the Covid survey, and thus, data from wave 9 main survey was used. Highest educational qualification was categorised as degree (International Standard Classification of Education-ISCED level 6), higher education below degree (ISCED level 4 and 5), A-level (ISCED level 3), O-level (ISCED level 2), and lower than O-level. Occupational class was measured by the National Statistics Socio-economic Classification (NS-SEC) five-class version (management & professional, intermediate, small employers & own account, lower supervisory & technical, and semi-routine & routine). Those who were not working in January or February, their occupational class was coded as 'not working in Jan/Feb'. Those who were working in January or February but not in wave 9 (and thus have no information of occupational class) were coded as 'not working in wave 9 only'. April wave only measured household earnings (i.e., earnings from paid work or self-employment), but not household income (e.g. pension among those not working). Therefore, quintiles of total household net income from wave 9 main survey were used.

## Statistical analyses

We did two analyses using the April wave and May wave, separately. All the analyses were stratified by gender as we are interested in the gender differences, and our pooled analyses showed that gender was an effect modifier in the association between unpaid care work and GHQ (p<0.01).

The association between unpaid care work and GHQ Likert scores was assessed by linear regression models, as GHQ Likert scores are normally distributed in the sample. All the regression model analyses were weighted (Stata command 'svyset') to take account of cross-sectional probability weight, clustering (primary sampling unit) and stratification (strata) at wave 9 main survey. This will provide estimates that are representative of the population of all adults (16+) resident in private households in the UK at the time of wave 9 main survey.

We also tested whether living with a partner is an effect modifier (testing Hypothesis 4) using an interaction term between partnership status and unpaid care variables, and stratified analyses were conducted when the *p*-value for the interaction term was lower than 0.05.

**Sensitivity analyses.** We conducted a sensitivity analysis to investigate the extent to which gender differences in unpaid care work remained after adjusting for demographic differences (same covariates as in the main analyses). Linear regression was used for the number of housework and childcare hours, and logistic regression was used for whether parents reduced or

adapted working hours in response to childcare or homeschooling. Three models are presented: an unadjusted model, a model adjusted for covariates but without baseline work hours, and a fully adjusted model. We also conducted a sensitivity analysis to test the potential 'actor-partner' effects [17, 18] in the association between unpaid care work hours (housework and childcare) and psychological distress amongst couples, that is to investigate whether partner's housework and childcare hours were related to the individual's distress in addition to their own. To test this 'actor-partner' effect, we limited the sample to couples and included partner's and actor's hours of housework or childcare in the model as well.

## Results

### Descriptive results

Table 1 shows sample characteristics by gender and by wave. The April and May samples have very similar characteristics. Compared to women, men were older and were more likely to live

**Table 1. Descriptive characteristics of men and women[a].**

| | April wave | | May wave | |
|---|---|---|---|---|
| | **Men** | **Women** | **Men** | **Women** |
| | **N = 6419** | **N = 9007** | **N = 5859** | **N = 8291** |
| | % | % | % | % |
| **Age-group** | | | | |
| 16/25 | 5.9 | 8.4 | 4.5 | 7.5 |
| 25/34 | 9.0 | 11.4 | 8.2 | 10.6 |
| 35/44 | 14.1 | 16.2 | 13.6 | 15.7 |
| 45/54 | 20.1 | 20.6 | 19.9 | 20.7 |
| 55/64 | 21.8 | 20.7 | 22.4 | 21.3 |
| 65/74 | 20.0 | 16.4 | 21.6 | 17.5 |
| 75 + | 9.2 | 6.2 | 9.9 | 6.7 |
| **Ethnicity** | | | | |
| White | 83.6 | 82.9 | 84.9 | 83.7 |
| Other Asian | 6.1 | 5.7 | 5.9 | 5.9 |
| African/Black/Caribbean | 1.9 | 2.6 | 2.0 | 2.6 |
| Other/mixed | 1.2 | 1.1 | 1.0 | 1.1 |
| Missing | 7.3 | 7.7 | 6.1 | 6.7 |
| **Living with a partner** | | | | |
| Yes | 77.7 | 67.6 | 76.0 | 66.0 |
| No | 22.3 | 32.4 | 24.0 | 34.0 |
| **Number of children aged 0–4** | | | | |
| 0 | 92.4 | 90.3 | 92.4 | 91.6 |
| 1 | 6.6 | 7.6 | 6.1 | 6.7 |
| 2+ | 2.0 | 2.0 | 1.6 | 1.8 |
| **Number of children aged 5–15** | | | | |
| 0 | 78.9 | 75.6 | 79.8 | 76.8 |
| 1 | 11.2 | 12.9 | 11.0 | 12.3 |
| 2+ | 9.9 | 11.5 | 9.2 | 10.8 |
| **Number of children aged 16–18** | | | | |
| 0 | 90.7 | 89.5 | 92.1 | 90.5 |
| 1 | 8.4 | 9.3 | 7.5 | 8.7 |
| 2+ | 0.8 | 1.2 | 0.5 | 0.8 |

*(Continued)*

**Table 1.** (Continued)

| | April wave | | May wave | |
|---|---|---|---|---|
| | **Men** | **Women** | **Men** | **Women** |
| | **N = 6419** | **N = 9007** | **N = 5859** | **N = 8291** |
| | % | % | % | % |
| **Baseline working hours** | | | | |
| Not working | 37.2 | 38.0 | 38.7 | 38.9 |
| Working PT | 7.6 | 23.0 | 7.8 | 22.5 |
| Working FT | 40.1 | 33.6 | 38.4 | 33.0 |
| Working FT with long hours | 15.0 | 5.3 | 14.9 | 5.4 |
| Missing | 0.1 | 0.2 | 0.1 | 0.1 |
| **Qualifications** | | | | |
| Degree | 34.2 | 31.0 | 34.9 | 31.8 |
| Higher education below degree | 11.1 | 14.8 | 11.5 | 14.8 |
| A-level | 10.9 | 10.7 | 10.5 | 10.7 |
| O-level | 19.0 | 20.1 | 19.0 | 19.9 |
| Lower | 16.8 | 15.0 | 17.3 | 15.4 |
| Missing | 8.0 | 8.4 | 6.8 | 7.4 |
| **Occupational class** | | | | |
| Management & professional | 28.5 | 24.6 | 28.1 | 25.1 |
| Intermediate | 5.2 | 9.2 | 4.8 | 9.2 |
| Small employers & own account | 5.7 | 3.4 | 5.8 | 3.4 |
| Lower supervisory & technical | 4.5 | 1.8 | 4.4 | 1.8 |
| Semi-routine & routine | 8.6 | 11.4 | 8.5 | 10.8 |
| Not working in Jan/Feb | 37.2 | 38.0 | 38.7 | 38.9 |
| Not working in wave 9 only | 7.7 | 9.4 | 7.0 | 8.6 |
| Missing | 2.6 | 2.2 | 2.6 | 2.2 |
| **Household income** | | | | |
| Lowest | 16.9 | 20.3 | 17.1 | 20.4 |
| 2 | 19.2 | 18.5 | 19.5 | 18.5 |
| 3 | 19.4 | 18.9 | 19.6 | 18.8 |
| 4 | 20.2 | 18.1 | 20.2 | 18.2 |
| Highest | 20.1 | 18.1 | 19.7 | 18.7 |
| Missing | 4.3 | 6.1 | 4.1 | 5.4 |
| **Mean GHQ during lockdown (SD)** | 11.22 (5.42) | 13.32 (6.30) | 11.27 (5.49) | 13.02(6.15) |
| **Mean GHQ at wave 9 (SD)** | 10.38 (4.97) | 11.61(5.62) | 10.01 (5.44) | 11.27 (5.97) |
| % missing | 8.7 | 9.0 | 5.9 | 6.6 |

[a] N is based on sample who has both GHQ and housework data at Covid survey and data are weighted using wave 9 survey weights.

with a partner, although they were slightly less likely to have a child in the household. Men were also more likely to work full-time or work full-time with long hours before the pandemic, were more likely to have a degree qualification, to be in a management & professional occupational class, to have higher household income and to have lower baseline GHQ score (i.e., better mental health) than women. The distributions of ethnicity are the same between men and women.

Table 2 shows the gender division of unpaid care work in April and May. On average, women spent about 15 hours per week doing housework in April and May, while men spent less than 10 hours per week on doing housework. Regarding childcare, women spent on

**Table 2. Descriptive characteristics of gender division of unpaid care work[a].**

| | April wave | | May wave | |
|---|---|---|---|---|
| | **Men** | **Women** | **Men** | **Women** |
| *Individual-level unpaid care* | | | | |
| Housework hours per week (mean) | 9.91 | 14.92 | 9.55 | 14.87 |
| Childcare/homeschooling hours per week (mean) | 12.03 | 20.54 | 11.61 | 22.47 |
| Reduce employment hours due to childcare/homeschooling | -- | -- | 11.6% | 16.8% |
| Adapted work patterns due to childcare/homeschooling | -- | -- | 29.5% | 36.7% |
| *Couple-level unpaid care* | **April wave** | | **May wave** | |
| Woman's share of housework | 63.5% | | 64.1% | |
| Mother's share of childcare/homeschooling | 62.2% | | 63.1% | |
| Reduce employment hours due to childcare/homeschooling | | | | |
| Neither | -- | | 64.9% | |
| Both | -- | | 3.5% | |
| Mother only | -- | | 21.1% | |
| Father only | -- | | 10.6% | |
| Adapted work patterns due to childcare/homeschooling | | | | |
| Neither | -- | | 35.4% | |
| Both | -- | | 14.2% | |
| Mother only | -- | | 32.4% | |
| Father only | -- | | 18.0% | |

[a] N is based on sample who has both GHQ and unpaid work variable at Covid survey and data are weighted using wave 9 survey weights.

average 20.5 hours per week on childcare/homeschooling in April, and this number increased to 22.5 hours per week in May. Men spent about 12 hours per week on childcare/homeschooling in April and May. Because of the time spent on doing childcare/homeschooling, one in six working mothers reduced their employment hours and one in three working mothers adapted their work patterns. Working fathers were 5 percentage points less likely to reduce working hours and 7 percentage points less likely to adapt work patterns due to childcare/homeschooling than working mothers.

Within couples, women shared 64% of housework and 63% of childcare, and it was more likely to be the mother than the father who reduced working hours (21% mother only vs. 11% father only) or changed employment schedules (32% mother only vs. 18% father only). Only 4% of couple parents both reduced their employment hours, while many more couple parents (14%) both adapted their work patterns because of childcare/homeschooling.

## Regression results

Table 3 shows the association between unpaid care work and GHQ for men and women in April in fully adjusted models. It shows that increased housework hours and childcare/homeschooling hours were associated with higher levels of psychological distress among women only. Among women, every one-hour increase in housework hours per week was associated with 0.05 (95%CI: 0.019, 0.071; $p = 0.001$) higher scores on the 36-point scale of GHQ, and every one-hour increase in childcare/homeschooling hours per week was associated with 0.02 higher scores of GHQ (95%CI: 0.006, 0.037; $p = 0.006$), which was a relatively weak association. No significant association was found among men. Within couples, women's share of involvement in housework and childcare/homeschooling was associated with neither men's nor women's GHQ.

**Table 3. Association between gender division of unpaid care work and GHQ in April wave[a].**

| | GHQ Likert, Men | | | GHQ Likert, Women | | |
|---|---|---|---|---|---|---|
| | Coef. (95%CI) | p-value | N | Coef. (95%CI) | p-value | N |
| *Individual-level unpaid care* | | | | | | |
| Housework hours per week | 0.024 (-0.005, 0.052) | 0.112 | 5541 | **0.045** (0.019, 0.071) | 0.001 | 7677 |
| Childcare/homeschooling hours per week | 0.022 (-0.009, 0.053) | 0.160 | 1582 | **0.022** (0.006, 0.037) | 0.006 | 2592 |
| *Couple-level unpaid care* | | | | | | |
| Woman's share of housework | 0.003 (-0.009, 0.015) | 0.629 | 3129 | 0.004 (-0.014, 0.021) | 0.685 | 3880 |
| Mother's share of childcare/homeschooling | -0.001 (-0.019, 0.016) | 0.936 | 849 | 0.004 (-0.013, 0.020) | 0.765 | 883 |

[a] Adjusted for age, ethnicity, living with a partner, number of children in the household by children's age group, baseline working hours, qualifications, occupational class, and baseline GHQ. Coef. with p<0.05 are shown in bold.

Table 4 shows the fully adjusted association between unpaid care work and GHQ for men and women in May. Similar to the April results, increased childcare/homeschooling hours were associated with higher levels of psychological distress among women only (coef. = 0.018, 95%CI: 0.001,0.034). Again, the strength of this association was relatively weak, as every one-hour increase in housework hours per week was only associated with 0.018 higher scores on the 36-point scale of GHQ. The association between housework hours and GHQ no longer

**Table 4. Association between gender division of unpaid care work and GHQ in May wave[a].**

| | GHQ Likert, Men | | | GHQ Likert, Women | | |
|---|---|---|---|---|---|---|
| | Coef.(95%CI) | p-value | N | Coef.(95%CI) | p-value | N |
| *Individual-level unpaid care* | | | | | | |
| Housework hours per week | 0.026 (-0.006, 0.057) | 0.116 | 5202 | 0.013 (-0.008, 0.034) | 0.234 | 7270 |
| Childcare/homeschooling hours per week | 0.016 (-0.009, 0.041) | 0.206 | 1417 | **0.018** (0.001, 0.034) | 0.036 | 2302 |
| Reduced employment hours due to childcare/homeschooling | | | | | | |
| No | ref | | 1252 | ref | | 1738 |
| Yes | 1.342 (-0.378, 3.061) | 0.126 | | 0.655 (-0.423, 1.734) | 0.233 | |
| Adapted work patterns due to childcare/homeschooling | | | 1250 | | | 1733 |
| No | ref | | | ref | | |
| Yes | **1.155** (0.296, 2.015) | 0.009 | | **1.393** (0.403, 2.382) | 0.006 | |
| *Couple-level unpaid care* | | | | | | |
| Woman's share of housework | -0.005 (-0.021, 0.011) | 0.524 | 2814 | -0.011 (-0.027, 0.005) | 0.185 | 2842 |
| Mother's share of childcare/homeschooling | 0.001 (-0.015, 0.018) | 0.880 | 766 | 0.015 (-0.003, 0.032) | 0.101 | 785 |
| Reduced employment hours due to childcare/homeschooling [b] | | | 752 | | | 820 |
| Neither | ref | | | ref | | |
| Both | -0.715 (-2.250,0.818) | 0.358 | | 0.113 (-2.794, 3.020) | 0.939 | |
| Mother only | 0.731 (-0.471, 1.933) | 0.231 | | 0.812 (-0.256, 1.880) | 0.136 | |
| Father only | **2.913** (1.321, 4.505) | <0.001 | | 0.001 (-2.269, 2.271) | 0.999 | |
| Adapted work patterns due to childcare/homeschooling | | | 752 | | | 820 |
| Neither | ref | | | ref | | |
| Both | 0.814 (-0.358, 1.985) | 0.172 | | 0.726 (-0.743, 2.194) | 0.551 | |
| Mother only | 0.566 (-0.874, 2.006) | 0.439 | | **1.821** (0.669, 2.973) | 0.002 | |
| Father only | **2.484** (1.367, 3.601) | <0.001 | | 0.626 (-1.437, 2.689) | 0.551 | |

[a] Adjusted for age, ethnicity, living with a partner, number of children in the household by children's age group, baseline working hours, qualifications, occupational class, and baseline GHQ. Coef. with p<0.05 are shown in bold.

exist (coef. = 0.01, 95%CI: -0.008,0.034) in May, and women's share of involvement in house-work and childcare/homeschooling was associated with neither men's nor women's GHQ.

The May wave additionally measured whether employment hours were reduced or adapted because of childcare/homeschooling. Fathers and mothers who adapted their work patterns due to childcare/homeschooling had on average 1.16 (95%CI: 0.296, 2.015) and 1.39 (95%CI: 0.403, 2.382) higher GHQ scores than those who did not, respectively. This was not a very strong association; however, this association was much stronger if he or she was the only mem-ber in the household who adapted their work pattern. Fathers had 2.48 higher GHQ scores (95%CI: 1.367, 3.601) if he was the only member of the couple to adapt his work pattern to accommodate childcare and mothers had 1.82 higher GHQ scores (95%CI: 0.669, 2.973) if she was the only member of the couple to adapted her work pattern to accommodate childcare, compared with fathers and mothers in couples where neither adapted their work patterns. In terms of adjusting working hours to accommodate childcare, fathers were more likely to have a higher GHQ score if he was the only member in the household who reduced working hours due to childcare/homeschooling (coef. = 2.91 vs. neither reducing work hours) but the same was not true for mothers.

**Partnership differences.** Partnership moderated ($p$ for interaction term <0.05) the asso-ciation between adapting work patterns and GHQ among women. Therefore, Table 5 shows the results stratified by lone mothers and couple mothers. Lone mothers who adapted work patterns due to childcare/ homeschooling had on average 3.93 higher GHQ scores (95% CI: 1.639, 6.223; $p$ = 0.001) than lone mothers who did not adapt work patterns. Adapting work patterns due to childcare/ homeschooling was not associated with couple mother's GHQ.

**Results from sensitivity analyses.** Sensitivity analysis (S2 and S3 Tables) shows that, after accounting for the demographic differences, the gender difference in childcare/homeschooling hours remains but is reduced (from a 9-hour gender difference in the unadjusted model to a 6-hour difference in the fully adjusted model in both April and May.) However, the gender dif-ferences in housework hours or work adaptation of working hours were not attenuated after accounting for demographic differences or baseline working hours. Results of the 'actor-part-ner' models are shown in S4 Table, with no significant partner effects found.

## Discussion

Using a large, nationally representative study of UK adults, this study assessed the gender divi-sion of unpaid care work at both individual and couple level during UK lockdown and tested how it was associated with changes in levels of psychological distress pre- and post-lockdown for men and women.

We hypothesized that there were continued gender differences in the unpaid care work during first lockdown and how have couples shared this unpaid care work between them. We start by providing evidence that, on average, women spent much more time doing housework

**Table 5. Association between adapting work patterns and GHQ stratified by couple mothers and lone mothers[a].**

|  | GHQ Likert, couple mothers | | | GHQ Likert, lone mothers | | |
|---|---|---|---|---|---|---|
|  | Coef. (95%CI) | *p*-value | N | Coef. (95%CI) | *p*-value | N |
| Adapted work patterns due to childcare/homeschooling |  |  |  |  |  |  |
| No | ref |  |  | ref |  |  |
| Yes | 0.962 (-0.073, 1.997) | 0.068 | 1413 | **3.931** (1.639,6.223) | 0.001 | 320 |

[a] Adjusted for age, ethnicity, number of children in the household by children's age group, baseline working hours, qualifications, occupational class, and baseline GHQ. Coef. with p<0.05 are shown in bold.

and childcare than men during lockdown, and women were more likely than men to reduce working hours and to adapt employment schedules due to increased time on unpaid care (Hypothesis 1 is supported). Within couples, women's share of unpaid work was as much as 64%. Our results suggest that the work schedules of working parents were widely affected by lockdown, as one-third of parents in this study adapted their work patterns because of childcare/homeschooling. Fewer parents reduced their working hours due to childcare/homeschooling, and it is likely that many parents did not feel able to reduce working hours in the context of potential increases in unemployment and redundancy during the crisis [29]. Within couples, where an adjustment in employment occurred, it was more likely to be the mother than the father who was the only member of the couple to reduce working hours and to change work schedules to accommodate childcare (Hypothesis 2 is supported). Our study suggests that the Covid-19 crisis did not force trends of gender convergence on unpaid work, and our result is consistent with previous research showing the continued gender inequality in divisions of unpaid care work among UK couples before the crisis [20].

Our next two hypotheses stated that the demands of unpaid care work during lockdown and the ways in which couples divided these demands, were associated with psychological distress during lockdown, and these associations were stronger for women than for men. These two hypotheses (4 and 5) were partly supported by our results. We found that women who spent long hours on housework were more likely to report increased levels of psychological distress in April, suggesting that women may be more affected by the increased responsibilities at home during the beginning of lockdown in terms of mental health. However, this association no longer exist in May. This is probably because some women exited employment or reduced their working hours over lockdown to accommodate the increased responsibilities at home, and recent evidence suggested that women were more likely to leave their jobs during lockdown than men [21]. We also found that more hours spent on childcare were significantly associated with increased levels of psychological distress for women only, but the association was relatively weak.

Among working parents, we found that fathers and mothers who adapted their work patterns to accommodate increased childcare/homeschooling hours had higher levels of psychological distress than those who did not, and the association is slightly stronger for mothers than for fathers. As home-schooling and childcare spillover into working hours, if reducing working hours is not feasible, adapting work patterns, such as working late into the evenings, early mornings or weekends, became an option for many parents. This may lead to long days, reduced sleep and time for rest or relaxation [30, 31], lack of physical exercise, feelings of loneliness and being overwhelmed by trying to meet work deadlines alongside family responsibilities, ultimately leading to psychological distressed [32]. In contrast, working parents who did not make such adjustment in their work may have other sources of support, such as a partner or parents to share the unpaid care, which allowed them to keep their original employment schedules.

Our research shows that levels of psychological distress are particularly high if he or she was the only member in the household who adapted work patterns. According to equity theory, couples evaluate both their contributions to the relationship and benefits from the relationship [13]. Unpaid care work has been depicted as a less prestigious activity that is also physically demanding and isolating [33, 34], especially during lockdown. Thus, if one partner is doing the bulk of unpaid care work, this may engender feelings of inequity and injustice in a relationship, and potentially impact psychological distress [35]. However, we found that the share of housework and childcare within couples was not associated with psychological distressed, indicating that the perceptions of equity or inequity in a relationship may be more important when people try to meet demands of multiple roles, such as balancing work and child care.

Our data does not have direct measures of perceptions of equality or fairness between couples; future research should more fully investigate the complexity of unpaid care work and its relationships with paid work, including concepts such as fairness.

The last research question was to test, among working parents, whether the above associations differ for those living with a partner compared with those who are not. We found that living with a partner provides a significant buffer in the association between adapting work patterns and psychological distress among women. Adapting one's work patterns due to childcare/ homeschooling seems to be more harmful for lone mothers than for couple mothers (Hypothesis 5 is partly supported). Without the support of a partner in the household during the height of the first wave of the Covid-19 pandemic, working lone parents were more likely to experience the conflicts of meeting several needs simultaneously, including childcare, homeschooling, housework, paid employment, and self-care [33]. Compared with couple mothers, lone mothers who change their work patterns to cope with childcare/ homeschooling are likely to have less support than partnered parents who are doing so. As a result, they may be working longer hours (when paid and unpaid work are combined) and feel the pressure of shouldering all of the work and family responsibilities themselves, thus contributing to psychological distress. Partner support is an important buffer for job-related stress and positive spill-over between family and work, such as be able to talk through difficulties at work may be important for people to recover from a stressful working day [16].

Many working lone mothers already struggled to combine work and family responsibilities before the pandemic [25]. The shut-down of formal childcare and rules of social distance mean that lone mothers juggle the pressure of being the sole breadwinner and child carer which seems to be particularly linked with high levels of psychological distress. Increased responsibilities at home during lockdown have made it even harder for lone mothers to continue working and this may have knock-ons for their return to work or further hardship as they try to juggle uncertain times ahead.

## Strengths and limitations

In this study, a large, nationally representative sample of men and women from across the entire adult age range were analysed and we were able to account for a number of important covariates using the information from the wave 9 main survey. We included several detailed measures of unpaid care, at both individual and couple level. However, this study has several limitations. First, our analyses did not assess the change in unpaid care work between pre-lockdown and lockdown and how this change has affected psychological distress. This is because childcare and housework hours were last measured in wave 8 (2016/18), which is 2 to 4 years prior to the lockdown, and thus, are less informative for the pre-lockdown level of unpaid work. In addition, childcare was measured by who is the main childcare giver rather than the specific hours in wave 8, making it difficult to compare the change of childcare between pre-lockdown and lockdown. Therefore, our study cannot estimate the causal effect of gender divisions of unpaid work on people's mental health, but we have adjusted for wave 9 GHQ before the pandemic to examine changes in GHQ scores and thus largely reduce the bias caused by reverse causality. Second, the Covid-19 web surveys have relatively lower response rates (at about 40%) than the main annual survey, and participants who do not use the internet were not included in the sample. It is possible that those were most affected by the lockdown did not participate in the Covid-19 web surveys, and thus our results may be underestimated. In addition, when assessing the couple-level division of domestic labour, our sample is limited to those who are living with a couple in the interview and both members having answered the relevant unpaid care questions. So, we were not able to measure the couple-level division of

unpaid care if only one member of the couple has participated in the survey, leading to selection bias as well. Besides, we conducted a 'complete case analysis', and thus, our results might be biased due to missing data. The largest percentage of missing is from covariates which were mostly measured in wave 9, as some participants did not participate in wave 9. We did not conduct a multiple imputation of missing data, as imputing missing data for those who did not participate in the interview may introduce more bias than the complete case analysis. Last, previous research has shown that men tend to overreport the amount of time and effort they allocate to childcare and homemaking activities [36]. In our study, hours spent on housework and childcare are self-reported, which might underestimate women's share of housework in the couple.

## Conclusions

Our study contributes to the growing literature corpus on the social consequences of the Covid-19 pandemic by focusing on possible unforeseen consequences of school closures and lockdown measures on men and women's psychological well-being, highlighting how such consequences might differ by gender and family structure. Our research suggests that juggling home working with homeschooling and childcare as well as extra housework is likely to lead to long-hour days and working non-standard patterns for many parents, and especially for lone parents. This has put a strain on parents and influence their mental health. Action is needed to better protect lone mothers and their children during the Covid-19 crisis, particularly given the resurgence of cases in many countries leading to on-going lockdowns. In the UK, schools are once again closed at the time of writing, with little public discourse regarding the difficulties this represents for lone mothers in particular. Our results suggest it is vital that governments and employers consider greater flexibility and support for lone mothers during the pandemic. In addition, continued gender inequality in divisions of unpaid care work during lockdown may put women at a greater risk of psychological distress. Awareness of continued gender biases in divisions of labour and their impact on psychological health is important for both couples and employers going forward.

## Supporting information

**S1 Table. Sample types for analysis.**
(DOCX)

**S2 Table. Gender differences in individual-level unpaid care work after adjusting for demographic differences in April wave.**
(DOCX)

**S3 Table. Gender differences in individual-level unpaid care work after adjusting for demographic differences in May wave.**
(DOCX)

**S4 Table. Actor-partner effects in the association between unpaid care and psychological distress amongst couples.**
(DOCX)

## Acknowledgments

We thank all participants in the Understanding Society study, Understanding Society researchers, and supporting staff who made the study possible.

## Author Contributions

**Conceptualization:** Baowen Xue, Anne McMunn.

**Data curation:** Baowen Xue, Anne McMunn.

**Formal analysis:** Baowen Xue.

**Funding acquisition:** Anne McMunn.

**Investigation:** Anne McMunn.

**Methodology:** Baowen Xue, Anne McMunn.

**Software:** Baowen Xue.

**Supervision:** Anne McMunn.

**Validation:** Anne McMunn.

**Writing – original draft:** Baowen Xue.

**Writing – review & editing:** Baowen Xue, Anne McMunn.

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
