## [Decision Letter · Decision Letter 0]

11 Jan 2021

PONE-D-20-35127

Gender differences in unpaid care work and psychological distress in the UK Covid-19 lockdown

PLOS ONE

Dear Dr. Xue,

Thank you for submitting your manuscript to PLOS ONE. After careful consideration, we feel that it has merit but does not fully meet PLOS ONE’s publication criteria as it currently stands. Therefore, we invite you to submit a revised version of the manuscript that addresses the points raised during the review process.

We look forward to receiving your revised manuscript.

Kind regards,

Thach Duc Tran, M.Sc., Ph.D.

Academic Editor

PLOS ONE

Journal Requirements:

Reviewers' comments:

Reviewer's Responses to Questions

**Comments to the Author**

1. Is the manuscript technically sound, and do the data support the conclusions?

Reviewer #1: Yes

Reviewer #2: Yes

2. Has the statistical analysis been performed appropriately and rigorously? 

Reviewer #1: Yes

Reviewer #2: Yes

3. Have the authors made all data underlying the findings in their manuscript fully available?

Reviewer #1: Yes

Reviewer #2: Yes

4. Is the manuscript presented in an intelligible fashion and written in standard English?

Reviewer #1: Yes

Reviewer #2: Yes

5. Review Comments to the Author

Reviewer #1: This article tackles important questions about the impact of Covid-19-related lockdowns on the ways household and childcare labor is handled within households and the effects of that division of labor on the mental health of women and men. Although they do not set out specifically to test a theory, the authors frame their research using role strain theory, arguing that it reasonable to assume that the necessity for parents to devote more time to domestic work, childcare, and remote learning while often also working from home may lead to increased role overload and inter-role conflict. These may lead, in turn, to increased psychological distress.

The study relies on a large sample of respondents from an established nationally representative survey (Understanding Society) who were followed up using online questionnaires during April and May of 2019. The sample of respondents is broken down into subsamples: all participants, couples, parents, couple parents, working, parents, and working couple parents. This is an appropriate and useful data set, although, as noted in the article’s discussion of limitations, it may be that some people who were most strongly impacted by the changes enforced by the pandemic did not have easy internet access, and so may have been least likely to respond.

The authors provide a good description of the measures used and of their limitations. They note, for example, that they could not measure changes in hours of unpaid work from pre- to post- the onset of the pandemic, because the last pre-Covid survey was done in 2016/18. They also note that hours of work are measure via self-report, which, besides general accuracy issues, presents something of a problem for the measurement of gender differences, in that men tend to overestimate their hours of work. However, since their findings indicate much higher reported hours of unpaid work for women than for men, the latter issue does not appear to undercut their results.

The descriptive findings show that, during the lockdown, women spent much more time on unpaid care work than men did, and that women were considerably more likely than men to adjust their employment hours to meet the demands of childcare. The gender differences in these descriptive results are not adjusted for other differences between women and men in the sample: men were older, more likely to live with a partner, slightly less likely to have a child in their household, and much more likely than women to work full-time or full-time with long hours. While the male-female differences in hours of unpaid work are quite strikingly large, it would be interesting to see the extent to which such differences are tempered by these demographic differences, and the paper makes no attempt to do this.

On the other hand, in the linear regression analyses conducted to examine the links between unpaid care work and psychological distress, as measured by the GHQ, are appropriately adjusted for these possible confounding variables. These analyses show that for women, but not men, there is an association between hours of care work and psychological distress, and that working parents who adapted their work patterns showed more distress than those who did not—especially if the other partner did not make such an adjustment.

There are several issues that I think deserve more extensive treatment in the Discussion section of this paper. Why might working parents who adjusted their work hours feel more distressed than those who did not? Is it because those who did not adjust had other sources of support that allowed them to keep their original schedules? Why did lone mothers who adapted their work patterns show bigger increases in distress than did couple mothers who did the same? What impact may perceived equity within couples have had on psychological distress? The latter issue is raised in the Introduction and is mentioned briefly and unsatisfyingly in the Discussion, but is not actually explored. It is true that there is no data in this study that has direct bearing on these questions. However, the questions deserve exploration in the service of developing future research based on these interesting findings.

Reviewer #2: The authors are seeking to examine the role of unpaid care work on psychological distress during the COVID lockdown in the UK. This is clearly a timely and important topic. I think that the manuscript has a lot of positive features and with some revisions could make a substantial impact on the field.

First, the Introduction is well written – however, rather than emphasizing the role strain theory, I think that role spillover might be more relevant. The lockdown brings the work and home roles together as never before; thus, the spillover is obvious and immediate. I understand the authors are not focusing on any specific theoretical framework but rather seek to address research questions; however, the literature and theories on these issues is plentiful. I think it would behoove them to pose specific hypotheses as opposed to questions.

I was glad to see that the authors used a baseline measure of distress, working hours and employment pre-COVID; however, they did not use a baseline pre-COVID measure of childcare. I understand their limitation with this issue and appreciate their acknowledging it.

I was expecting to see partner effect analyses based on the discussion of crossover effects in the Introduction. I see that women’s housework share was entered into the couple-level analyses. However, I think it would be interesting to see if at the individual level partner’s housework hours/week and childcare hours/week were related to the individual’s distress in addition to the actor effects.

In the Discussion, the authors focus on the role strain theory but I am wondering if spillover might be more relevant here. Additionally, the authors should discuss long-term impacts of COVID lockdowns as the fall and winter have seen a resurgence of cases and social restrictions. It is one thing if this was an acute couple of months, but this has become chronic - how might the relationships with these constructs change?

Overall, I feel that the manuscript is timely but the authors need to do more in their analyses to examine the actor-partner effects.

6. PLOS authors have the option to publish the peer review history of their article (what does this mean?). If published, this will include your full peer review and any attached files.

Reviewer #1: No

Reviewer #2: No

---

## [Author Response · Author response to Decision Letter 0]

7 Feb 2021

Response to reviewers’ comments

Reviewer #1: This article tackles important questions about the impact of Covid-19-related lockdowns on the ways household and childcare labor is handled within households and the effects of that division of labor on the mental health of women and men. Although they do not set out specifically to test a theory, the authors frame their research using role strain theory, arguing that it reasonable to assume that the necessity for parents to devote more time to domestic work, childcare, and remote learning while often also working from home may lead to increased role overload and inter-role conflict. These may lead, in turn, to increased psychological distress. The study relies on a large sample of respondents from an established nationally representative survey (Understanding Society) who were followed up using online questionnaires during April and May of 2019. The sample of respondents is broken down into subsamples: all participants, couples, parents, couple parents, working, parents, and working couple parents. This is an appropriate and useful data set, although, as noted in the article’s discussion of limitations, it may be that some people who were most strongly impacted by the changes enforced by the pandemic did not have easy internet access, and so may have been least likely to respond.The authors provide a good description of the measures used and of their limitations. They note, for example, that they could not measure changes in hours of unpaid work from pre- to post- the onset of the pandemic, because the last pre-Covid survey was done in 2016/18. They also note that hours of work are measure via self-report, which, besides general accuracy issues, presents something of a problem for the measurement of gender differences, in that men tend to overestimate their hours of work. However, since their findings indicate much higher reported hours of unpaid work for women than for men, the latter issue does not appear to undercut their results.

Our response: We thank the reviewer for the positive comments. 

The descriptive findings show that, during the lockdown, women spent much more time on unpaid care work than men did, and that women were considerably more likely than men to adjust their employment hours to meet the demands of childcare. The gender differences in these descriptive results are not adjusted for other differences between women and men in the sample: men were older, more likely to live with a partner, slightly less likely to have a child in their household, and much more likely than women to work full-time or full-time with long hours. While the male-female differences in hours of unpaid work are quite strikingly large, it would be interesting to see the extent to which such differences are tempered by these demographic differences, and the paper makes no attempt to do this. On the other hand, in the linear regression analyses conducted to examine the links between unpaid care work and psychological distress, as measured by the GHQ, are appropriately adjusted for these possible confounding variables. These analyses show that for women, but not men, there is an association between hours of care work and psychological distress, and that working parents who adapted their work patterns showed more distress than those who did not—especially if the other partner did not make such an adjustment.

Our response: We have now investigated gender differences in unpaid care work after adjusting for demographic differences. Results are shown in S2 Table and S3 Table in the Supporting information. Three models are presented: an unadjusted model, a model adjusted for covariates but without baseline work hours (because we felt this may be an over adjustment as the ‘causal’ direction between employment and unpaid care work could go in both directions), and a fully adjusted model (same covariates as in the main analyses.) Both the coefficient for gender differences in the number of hours and the average marginal values for hours spent doing unpaid care work in the past week for men and women are shown in the tables. After accounting for the demographic differences, the gender difference in childcare/homeschooling hours remains but is reduced (changed from a 9-hour difference in the unadjusted model to a 6-hour difference in the fully adjusted model for April and May). However, the gender differences in housework hours or work adaptation of working hours were not attenuated after accounting for demographic differences or baseline working hours. Therefore, the results show that even after accounting for the demographic differences, women still spent much more time on unpaid care work than men during the first lockdown in the UK, and it was more likely to be the mother than the father who reduced working hours or changed employment schedules due to increased time on childcare. Relevant texts have been added to the Method (p 10 Sensitivity analyses) and Results sections (p 25 Results from sensitivity analyses).

There are several issues that I think deserve more extensive treatment in the Discussion section of this paper. Why might working parents who adjusted their work hours feel more distressed than those who did not? Is it because those who did not adjust had other sources of support that allowed them to keep their original schedules? Why did lone mothers who adapted their work patterns show bigger increases in distress than did couple mothers who did the same? 

Our response: We thank the reviewer for the useful comments on how to improve the discussion section. We have added text emphasising that parents who did not adjust their work patterns may have other sources of support that allowed them to keep their original schedules (p27 lines 385-388). We also have added text to explain possible reasons of why lone mothers who adapted their work patterns show bigger increases in distress than couple mothers who did the same (p28 lines 410-417), such as lone mothers have conflicts in meeting several needs at the same time without a partner’s support to buffer the stress of combining work and work adaption with childcare.

What impact may perceived equity within couples have had on psychological distress? The latter issue is raised in the Introduction and is mentioned briefly and unsatisfyingly in the Discussion, but is not actually explored. It is true that there is no data in this study that has direct bearing on these questions. However, the questions deserve exploration in the service of developing future research based on these interesting findings.

Our response: We thank the reviewer for the useful suggestion to address perceptions of equity within couples more thoroughly. We have added text on how perceived equity within couples may potentially impact psychological distress (p 27 lines 390-393), and highlight the need for future research on this topic (p27 line 398-401).

Reviewer #2: The authors are seeking to examine the role of unpaid care work on psychological distress during the COVID lockdown in the UK. This is clearly a timely and important topic. I think that the manuscript has a lot of positive features and with some revisions could make a substantial impact on the field.

First, the Introduction is well written – however, rather than emphasizing the role strain theory, I think that role spillover might be more relevant. The lockdown brings the work and home roles together as never before; thus, the spillover is obvious and immediate. I understand the authors are not focusing on any specific theoretical framework but rather seek to address research questions; however, the literature and theories on these issues is plentiful. I think it would behoove them to pose specific hypotheses as opposed to questions.

Our response: We thank the reviewer for the useful suggestions for our theoretical background and the use of hypotheses. We have added the role spillover theory in the Introduction (p 4, lines 84-94), and have posed specific hypotheses as opposed to questions (p6). In the discussion section, research questions have been replaced with hypotheses accordingly (p25-28). 

I was glad to see that the authors used a baseline measure of distress, working hours and employment pre-COVID; however, they did not use a baseline pre-COVID measure of childcare. I understand their limitation with this issue and appreciate their acknowledging it.

Our response: We thank the reviewer for the positive comments.

I was expecting to see partner effect analyses based on the discussion of crossover effects in the Introduction. I see that women’s housework share was entered into the couple-level analyses. However, I think it would be interesting to see if at the individual level partner’s housework hours/week and childcare hours/week were related to the individual’s distress in addition to the actor effects.

Our response: We have conducted a sensitivity analysis to investigate the actor-partner effect as suggested by the reviewer. Results are shown in S4 Table in the Supporting information. Women’s own high childcare/ homeschooling hours were associated with increased psychological distress in May, but no significant association with partner’s unpaid work hours was seen for women or men. Relevant texts were added to the Method (p 10 Sensitivity analyses) and Results sections (p 25 Results from sensitivity analyses).

In the Discussion, the authors focus on the role strain theory but I am wondering if spillover might be more relevant here. Additionally, the authors should discuss long-term impacts of COVID lockdowns as the fall and winter have seen a resurgence of cases and social restrictions. It is one thing if this was an acute couple of months, but this has become chronic - how might the relationships with these constructs change? Overall, I feel that the manuscript is timely but the authors need to do more in their analyses to examine the actor-partner effects.

Our response: We have added discussion of the spillover effect between family and work, both negative spillover (p27, lines 380-385) and positive spillover (p28, lines 414-417). We thank the reviewer for the important point regarding the long-term impacts of lockdowns which has indeed become the case with even greater restrictions in the UK currently. We have added text on p30 (lines 464-469) and hope to study the longer term experiences of lockdown going forward.

Response to journal requirements

Our response: We have carefully read the links and have made sure our manuscript meets PLOS ONE’s style requirements. 

Our response: We have added the following text in the Methods: 

“Participants gave informed oral consent to take part in each wave of the study and were enrolled only after consent was provided.”

Our response: We have corrected this statement. Our data are not available upon request; they are publicly available through the UK Data Service. Researchers who would like to use Understanding Society need to register with the UK Data Service before being allowed to apply for or download datasets. More information: 

https://www.understandingsociety.ac.uk/documentation/access-data

Our response: We have added the Data Availability statement in the cover letter.

Our response: We have included captions for Supporting Information files at the end of our manuscript, and have updated in-text citations to match accordingly.

---

## [Decision Letter · Decision Letter 1]

17 Feb 2021

Gender differences in unpaid care work and psychological distress in the UK Covid-19 lockdown

PONE-D-20-35127R1

Dear Dr. Xue,

We’re pleased to inform you that your manuscript has been judged scientifically suitable for publication and will be formally accepted for publication once it meets all outstanding technical requirements.

Kind regards,

Thach Duc Tran, M.Sc., Ph.D.

Academic Editor

PLOS ONE

Additional Editor Comments (optional):

Reviewers' comments:

Reviewer's Responses to Questions

**Comments to the Author**

1. If the authors have adequately addressed your comments raised in a previous round of review and you feel that this manuscript is now acceptable for publication, you may indicate that here to bypass the “Comments to the Author” section, enter your conflict of interest statement in the “Confidential to Editor” section, and submit your "Accept" recommendation.

Reviewer #1: All comments have been addressed

2. Is the manuscript technically sound, and do the data support the conclusions?

Reviewer #1: Yes

3. Has the statistical analysis been performed appropriately and rigorously? 

Reviewer #1: Yes

4. Have the authors made all data underlying the findings in their manuscript fully available?

Reviewer #1: Yes

5. Is the manuscript presented in an intelligible fashion and written in standard English?

Reviewer #1: Yes

6. Review Comments to the Author

Reviewer #1: (No Response)

7. PLOS authors have the option to publish the peer review history of their article (what does this mean?). If published, this will include your full peer review and any attached files.

Reviewer #1: No

---

## [Editor Report · Acceptance letter]

23 Feb 2021

PONE-D-20-35127R1 

Gender differences in unpaid care work and psychological distress in the UK Covid-19 lockdown 

Dear Dr. Xue:

I'm pleased to inform you that your manuscript has been deemed suitable for publication in PLOS ONE. Congratulations! Your manuscript is now with our production department. 

Kind regards, 

on behalf of

Dr. Thach Duc Tran 

Academic Editor

PLOS ONE